METHODS

# A method for analysing tissue motion and deformation during mammalian organogenesis

**Morena Raiola[1]\*, Isaac Esteban[1¤a], Kenzo Ivanovitch[1¤b], Miquel Sendra[1¤c], Miguel Torres[1,2]\***

**1** Cardiovascular Regeneration Program, Centro Nacional de Investigaciones Cardiovasculares (CNIC), Madrid, Spain, **2** Centro de Investigación Biomédica en Red de Enfermedades Cardiovasculares (CIBERCV), Madrid, Spain

¤a Current address: Institute for Biomedical Engineering, ETH Zürich, Zürich, Switzerland
¤b Current address: Developmental Biology of Birth Defects, Institute of Child Health, University College London, London, UK
¤c Current address: CNRS, UTLN, LIS 7020, Turing Centre for Living Systems, Aix Marseille University, Marseille, France
\* mtorres@cnic.es, mraiola@cnic.es

## Abstract

Understanding tissue morphogenesis is an important goal in developmental biology and tissue engineering. Accurately describing tissue deformation processes and how cell rearrangements contribute to these is a challenging task. Live analysis of morphogenesis in 3D is frequently used to obtain source data that allow to extract such features from developing organs. However, several limitations are encountered when applying these methodologies to mammalian embryos. The mouse embryo is the most frequently used model, but most studies use a very limited number of specimens and present only individual acquisitions due to constraints imposed by embryo culture and imaging. Here, we leverage live imaging of mouse heart development to build a novel computational framework that overcomes these limitations. Our methodology first extracts tissue dynamics from individual specimens and then integrates these fragmented datasets into a deterministic and dynamic consensus model of heart development. This integrated model allows us to quantify patterns of tissue growth and anisotropy and generate an in-silico fate map of cardiomyocyte trajectories. This work provides a foundational toolkit for dissecting the complex morphogenetic processes underlying mammalian organogenesis, converting collections of variable live images into robust, quantitative blueprints of development.

## Author summary

How can a small cluster of cells become a beating heart? This is a fascinating question, and one that science has not yet fully answered. Observing this process in real time is challenging: capturing video during embryonic development

**Data availability statement:** The authors confirm that all data underlying the findings are fully available without restriction. The authors declare that all data supporting the findings of this study are either included in the article and its supplementary information. Additionally, all raw and processed data generated during this study have been archived on a dedicated Mendeley Data server and can be accessed at the following addresses: • Mendeley Data: 10.17632/54gbvnsgnp.1 LINK: https://data.mendeley.com/datasets/54gbvnsgnp/1 • Code is available in github.com/MorRaiola/QuantitativeAnalisysOfHTMorphogenesis.git.

**Funding:** M.R. was a recipient of a Marie Skłodowska-Curie postdoctoral contract from H2020-MSCA-ITN-2016-722427. This work was funded by grants PGC2018-096486-B-I00 and PID2022-140058NB-C31 from the Agencia Estatal de Investigación to M.T.; Comunidad de Madrid grant P2022/BMD-7245 CARDIOBOOST-CM to M.T.; European Research Council AdG ref. 101142005 to M.T. The CNIC Unit of Microscopy and Dynamic Imaging is supported by FEDER 'Una manera de hacer Europa' (ReDIB ICTS infrastructure TRIMA@CNIC, MCIN). The CNIC is supported by the Instituto de Salud Carlos III (ISCIII), the Ministerio de Ciencia, Innovación y Universidades (MICIU) and the Pro CNIC Foundation, and is a Severo Ochoa Center of Excellence (grant CEX2020-001041-S funded by MICIU/AEI/10.13039/501100011033). The funders had no role in study design, data collection and analysis, decision to publish, or preparation of the manuscript.

**Competing interests:** The authors have declared that no competing interests exist.

requires dedicated techniques and many hours at the microscope. Often, researchers only obtain brief observation windows, too limited to cover the entire process. In practice, we usually end up with just scattered fragments of a much longer story.

In our work, we found a way to turn these fragments into a continuous narrative. Using live imaging of mouse embryos and computational tools, we were able to combine many partial observations into a single dynamic model of how the heart forms. It's a bit like creating a time-lapse: we can watch the tissue grow, the shape of the organ change, and even follow the paths of the cells that build it.

By bringing these segments together into a coherent blueprint, we offer a new perspective on how nature brings the heart to life. Beyond revealing part of this developmental mystery, our approach also opens new avenues for understanding how other complex organs form.

## Introduction

Recent advances in microscopy physics and the development of fluorescent proteins detectable in living cells have revolutionized *in vivo* observation of organogenesis in mammalian embryos [1–5]. This imaging process generates high-resolution spatio-temporal 3D+t data, providing a digital representation of organ morphogenesis at tissue and cellular scale. However, extracting biological meaning from these data necessitates *ad-hoc* image processing and data analysis techniques.

Characterizing the movement and deformation patterns of developing tissues poses a significant challenge in developmental biology. A large body of work has established that morphogenesis emerges from collective cell behaviours and the mechanical forces they generate [6,7]. Powerful computational frameworks have been developed to quantify these dynamics. For instance, in systems like the zebrafish embryo, where tissues can exhibit fluid-like behaviour, unsupervised analysis of Lagrangian trajectories from complete 4D datasets has been shown to reveal deformation patterns that match the embryonic fate map [8].

However, the direct application of such powerful Lagrangian frameworks to mammalian organogenesis is fundamentally constrained by the nature of the data that can be realistically acquired. Live imaging often falls short in capturing the full timeline of organ development or the entire region of interest within a single acquisition. To balance spatial and temporal resolution with field size in 3D+t imaging and enable cell tracking, data acquisition is typically limited to a few hours to avoid signal degradation and embryo damage. As a result, several acquisitions that cover different developmental stages have to be integrated to study the full developmental timeline [5].

Deep tissue imaging is further constrained by limited laser penetration in non-transparent specimens, such as mouse embryos, resulting in incomplete sampling. In addition, embryo movement during growth can also cause sample to drift out of focus, leading to the loss of peripheral regions of the organ of interest during

acquisition. Here, we developed a new methodology to cope with these challenges using early mouse heart development as a model. Although heart morphogenesis is a generally consistent and robust process that transforms the early heart primordium into a functional four-chambered organ, it exhibits significant variability in shape and maturation rates among individual embryos [5,9]. To address both the technical limitations of imaging and this inherent biological variability, we developed a systematic framework for the deterministic fusion of spatiotemporal data. Our method integrates multiple, incomplete live imaging datasets to quantify and compare myocardial motion and tissue deformation. The framework enables the analysis of tissue motion and deformation directly from individual live imaging datasets.

To synchronize these individual datasets, we developed a staging system that temporally maps our 3D live images to a previously described pseudodynamic Atlas of heart morphogenesis [9]. To account for variability in heart shape, we then implemented staging and warping processes that register each individual specimen onto the synchronous Atlas geometry. The Atlas thus serves as a common, deterministic coordinate system, a 'master timeline' and a 'master geometry', for data integration.

By projecting our individual motion profiles onto this unified spatiotemporal framework, we could deterministically reconstruct the cumulative tissue deformation and generate the first *in-silico* fate map of early heart morphogenesis. This high-fidelity, descriptive model allowed us to propose a new, data-driven model for mouse heart tube formation. Our methodology provides a robust framework that transforms highly variable, individual datasets into a unified and quantitative description of organogenesis, enabling a systematic analysis of the patterns and variability of tissue morphogenesis.

## Results

To quantify the deformation patterns and their variability during early mammalian heart development, we have developed a workflow that combines multiple live images in time and space. Our workflow, illustrated in Fig 1, consists of four key steps: (1) Estimating individual live image motion to describe heart tube shaping as a continuous process (blue shapes); (2) Integrating multiple live images into a consensus spatiotemporal reference by aligning each continuous motion profile with a published 3D + t Atlas [9] (red shapes); (3) Quantifying tissue deformation during early morphogenesis (yellow shapes); and (4) Creating an *in-silico* fate map to analyse heart tube morphogenesis at both pseudo-cellular and regional levels (green shapes). In the following sections, we will describe the results of each of these steps in sequence.

### Estimating individual live image motion

To capture the continuous motion of the developing heart tube, our analysis is based on a comprehensive dataset of 3D + t live images, including both published [5] and newly acquired data. The dataset includes 16 embryos at different stages of heart development, from the cardiac crescent to the linear heart tube (S1 Table). During this period the heart tissues undergo complex reorganization with a change in topology that converts a flat epithelial monolayer into a tube with two inflow (IFT) and outflow (OTF) tracts.

To visualize myocardial tissue and provide landmarks for validation, the imaged embryos were categorized based on their genetic reporters: some carried the Nkx2.5-GFP allele [10], while others had a combination of the Nkx2.5Cre allele [11] and *Rosa26* fluorescent reporter alleles (S1 Table). To facilitate cell tracking, Nkx2.5-GFP carriers also had cells labelled at random through the recombination of the Rosa26R Cre-reporter allele, induced by tamoxifen activation of CreERT2 from the RERT allele. Additional embryos were labelled with Mesp1Cre to report all mesodermal cells or Islet-1Cre to mark the second heart field.

Live imaging of these embryos generated high-resolution hyperstacks, capturing the cardiogenic domain over acquisitions lasting from two to over ten hours. These hyperstacks were stored in a 5D format (x, y, z, c, t) with a typical xy size of 1024x1024 pixels. Spatially, each frame had a fine in-plane resolution (e.g., 0.593 × 0.593 μm) and an axial z-step between 3 and 6 μm. Temporally, the interval between frames varied from 4 to 24 minutes depending on the specific specimen.

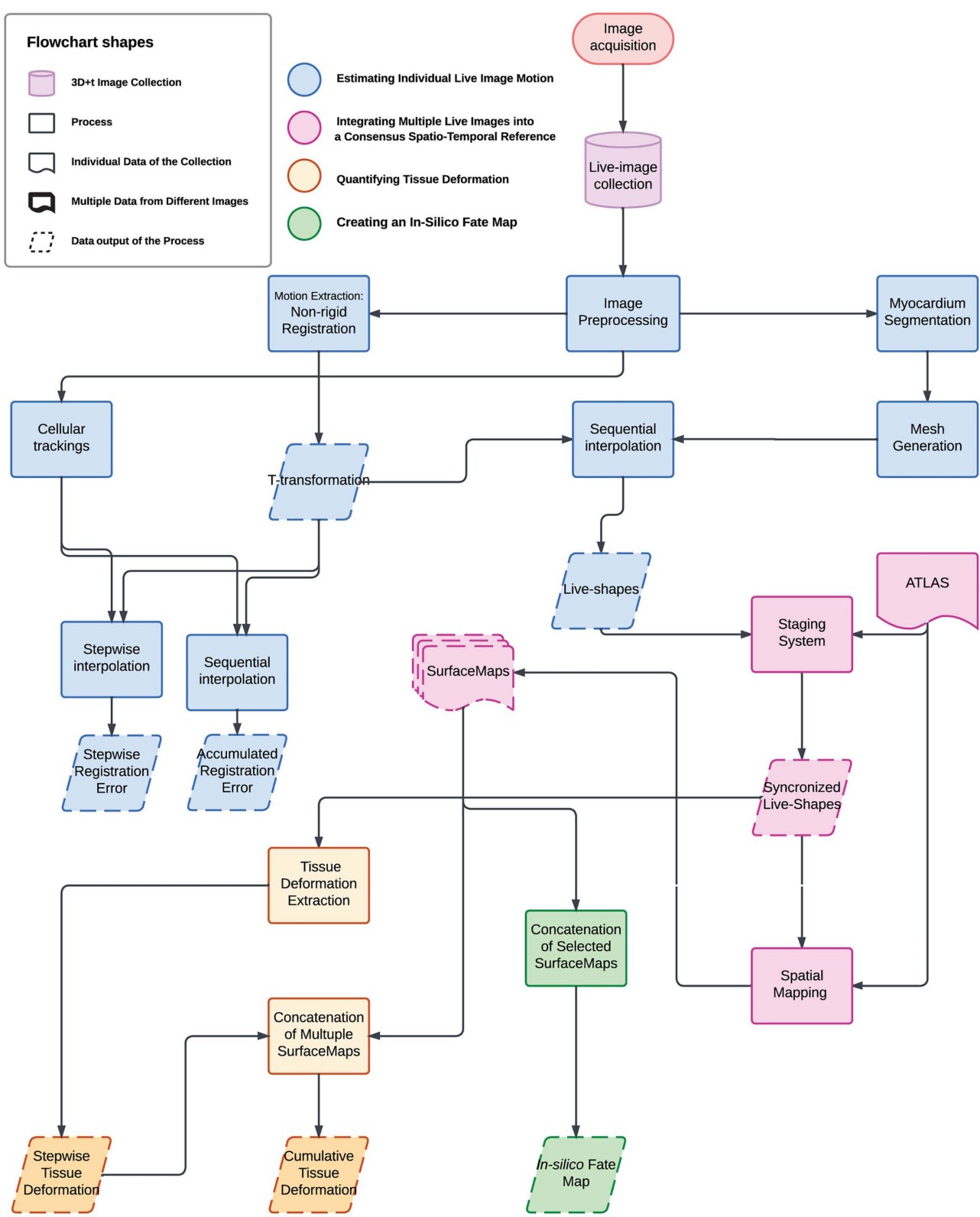

**Fig 1. Workflow to extract and define myocardial motion and deformation patterns during early heart morphogenesis.** A comprehensive pipeline for the 3D imaging of early heart morphogenesis. This figure provides an overview of the workflow used in this study. For full details of each methodological step, please refer to the MATERIALS AND METHODS section. The computational workflow comprises four main components: 1) Estimating

Individual Live Image Motion (blue shapes): Following image preprocessing, we extracted the underlying motion using a non-rigid registration algorithm, resulting in a set of Transformation. The accuracy of motion detection was evaluated by comparing tracked cells to computed tracking using a stepwise and sequential interpolation. The myocardium tissue was segmented at one time point, and a mesh was generated. By interpolating this mesh with the set of T-transformation, we derived 'Live-Shape', a continuous description of heart tissue motion. 2) Integrating Multiple Live Images into the Atlas (red shapes): Individual live image motions were integrated into a high-resolution Atlas. This involved a staging system to synchronize 'Live-Shape' sequences along the Atlas time reference and a spatial mapping strategy to project staged 'Live-Shape's into the Atlas spatial framework. The result was a set of SurfaceMap, representing the motion of each specimen within the Atlas. 3) Quantifying Tissue Deformation (yellow shapes): Individual tissue deformation patterns were extracted, mapped into the Atlas, and variability was assessed. We defined stepwise tissue deformation and cumulative deformation to quantify these changes over time. 4) Creating an In-Silico Fate Map (green shapes): An in-silico fate map of the myocardium was constructed for the developmental window between E7.75 and E8.25 by concatenating motion profiles, providing insights into the spatial and temporal dynamics of early heart morphogenesis. Our dataset includes multiple specimens raging from E7.75 to E8.25 (12 hours). Two different Rosa26Rtdtomato[+/-], Nkx-2.5eGFP specimens are aligned on a pseudo-timeline, representing the transition from the cardiac crescent to the linear heart tube.

Our first computational challenge was to extract the continuous tissue motion from these 4D image stacks. As a crucial preprocessing step, we first resampled the images to an isotropic voxel resolution, as described in MATERIALS AND METHODS - Image Preprocessing. We then applied a non-rigid registration technique, employing an adapted version of the MIRT algorithm [12] (see MATERIALS AND METHODS - MIRT algorithm). This intensity-based algorithm operates sequentially, aligning consecutive frames to extract the underlying transformation field (T) that describes the flow of the 3D intensity pattern between them (Fig 2A). The result of this process is a dense set of transformation fields, {Ti}, which provides a continuous, quantitative description of the tissue's movement and serves as the foundation for tracking any point throughout the video.

**Validating the motion estimation**

To determine the accuracy of the algorithm in estimating myocardial motion, we used the position of randomly labelled cells in the live images as ground truth. These cells were automatically tracked and then manually corrected using Imaris software version 9.5, resulting in a total of 9584 points/cell centroids derived from 9 embryos (see S2 Table). This included myocardial, splanchnic, and endodermal cells without distinction, labelled with either membrane-GFP or membrane-Tomato (Fig 2B, see also S2 Table). However, only non-dividing cells were considered for the validation process.

We created two test sets: the first set was used to estimate the intrinsic error of the registration algorithm when applied frame-by-frame (stepwise error); the second set was used to evaluate the error when the algorithm was implemented sequentially for more than two consecutive frames (accumulated error).

For the stepwise error, the test set was built applying the *morphSurf3D* function [13], between pairs of consecutive frames. The function determined the consecutive position (x', y', z') for each cell position (x, y, z) in the ground truth dataset (shown in blue/black in Fig 2C). For the accumulated error, the test set was generated by sequentially interpolating the initial cell positions (x, y, z) with the set of transformation fields {Ti}. The output of the first iteration was used as the input for the next iteration (shown in blue/pink in Fig 2C'). We measured the accuracy of the method by calculating the Eulerian distance between the position of the tracked cells (ground-truth) and the positions obtained from the registration process (test-set).

The results, shown in Fig 2D, demonstrate that the registration algorithm maintained a consistent stepwise error of approximately 2.5 μm throughout the video, well below the average cell diameter (~20 μm, see also Fig 2G). However, as the registration progressed, cumulative effects and interpolation approximations ({Ti}) caused the error to increase over time. This trend is illustrated by the pink line in Fig 2D, where the mean error for a 200minute sequence reached approximately 10 μm. To mitigate this issue, we introduced a bidirectional registration strategy, using the midpoint of the video (t N/2) as the starting point. This divided the process into two independent registration directions (Fig 2E). As shown by the orange line in Fig 2E', this approach successfully halved the mean accumulated error.

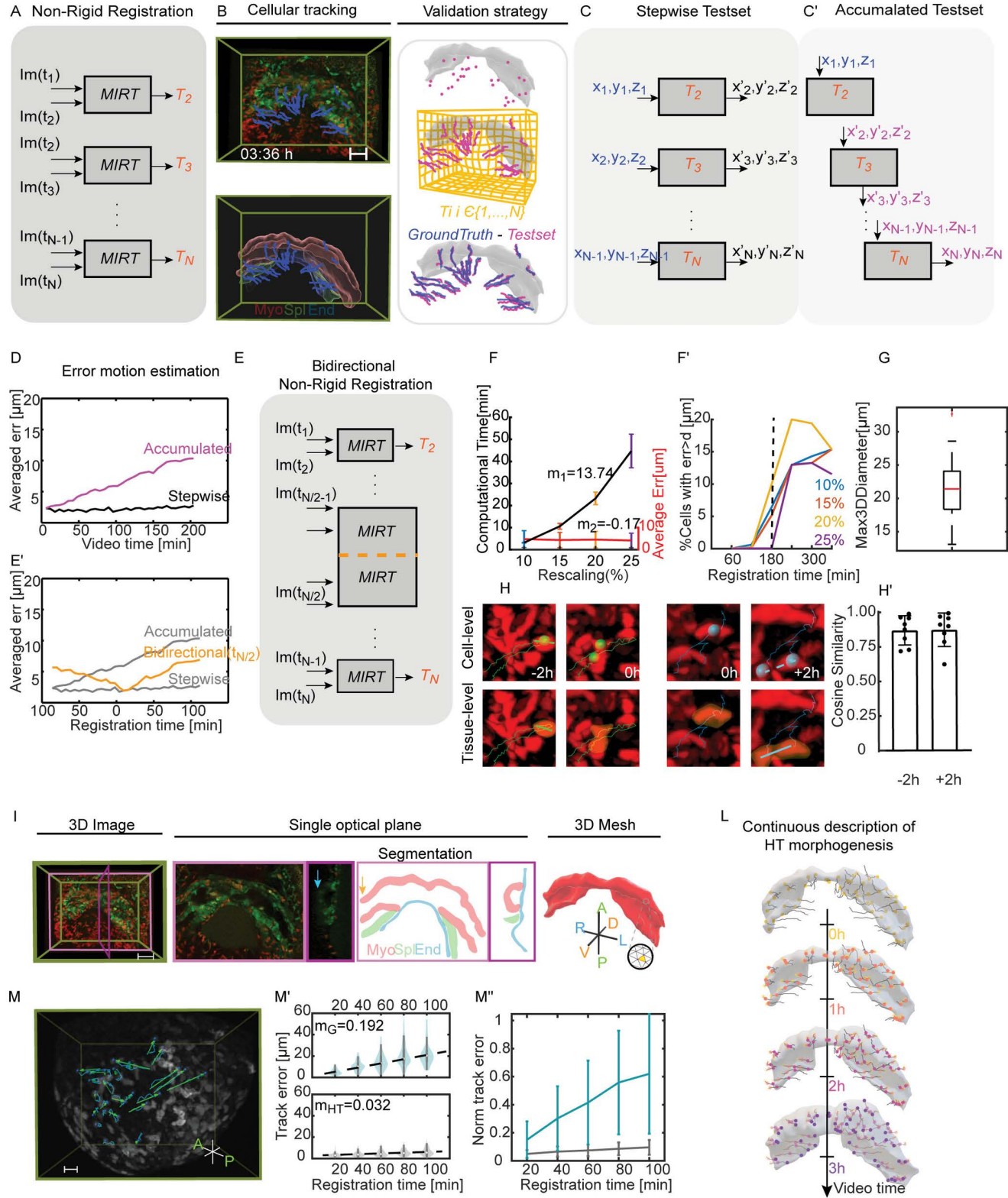

**Fig 2. Estimating individual live image motion to describe heart tube shaping as a continuous process.** (A), MIRT algorithm was applied frame-by-frame (I(ti)-I(ti+1)), extracting the set of deformation fields (T1, T2, …, TN) underlying the live images. (B), The top image illustrates the automatic cell

tracking (blue tracks) on a live image (Nkx2.5eGFP; Rosa26Rtdtomato+/-). The bottom image is an Imaris reconstruction of the HT and the relative automatic cell tracking. This tracking serves as the ground truth dataset. On the left, there is a representation of the validation strategy. By applying the deformation fields, we built our test set (pink lines). We computed the Eulerian distances between the ground truth and the test set to assess the accuracy of the registration method. The scale bar is 100 µm and the time is represented in [hh:mm]. (C), The graph illustrates the method for generating the stepwise test set. For each centroid in the ground truth dataset, the T-transformation determines its new position (x', y', z') at the next time point. (C'), The sequential test set is generated by sequentially interpolating the initial cell positions (x, y, z, t0) (shown in blue) using the transformation set {Ti}. The output from each iteration (shown in pink) serves as the input for the next iteration. (D), Average stepwise error in µm for a single embryo (e16) as a proof of concept. The black line represents the average stepwise error for 58 cells tracked over 204 minutes (35 frames), while the accumulated error is shown in pink. (E), Bidirectional approach started non-rigid registration at t(N/2), N = number of frames. (E'), Quantification of the accumulated error when the initial registration is fixed at t(N\2). (F), The comparison of average error and computation time trends of images rescaled at 10, 15, 20 and 25% of 3 embryos (e01, e02, e05). The analysis refers to the stepwise registration between only two consecutive frames. Values m1 and m2 represent the slope of the linear regressions of the two trends. (F'), The graph shows the percentage of points for which error between predicted and actual positions is greater than the mean diameter of a cell (d = 20 µm) for different rescaling rates (colour code). The dotted line represents the threshold we set to choose the resolution that best guarantees a balance between registration error and computation time. The threshold is set at 180 min of registration time. (G), Cardiomyocytes diameter is determined by calculating the maximum Feret diameter from the 3D segmentation of 26 cells. (H), The images provide a close-up view of cells under two different conditions: one where cells had already divided at time 0 (blue spots), and another where cells had not yet divided at time 0 (green spots). The orange mask outlines the segmented cell tissue at time 0 and after applying the T-transformation (t-2h; t+2h). Dotted lines illustrate the actual elongation of the tissue at t-2h and t+2h, while the straight line within the tissue indicates the direction of maximum elongation of the transformed segmentations. (H'), Cosine similarity was calculated between the actual tissue elongation and the elongation predicted by the T-transformation for 8 cells within the -2h to +2h interval. (I), 3D image of the HT of an Nkx2.5eGFP; Rosa26Rtdtomato+/- embryo(e02). The image includes a single ventral plane in a pink frame, and a single lateral plane with its segmentation section in a magenta frame. The myocardium (red), splanchnic mesoderm (green), and endoderm (blue) are visible in the segmentation section. The yellow arrow indicates the incomplete part of the right IFT. The blue arrow indicated the signal degradation in OFT. The segmentation was transformed into a volumetric mesh, defined as faces (yellow area), nodes (blue dots) and edges (orange sides). Scale bar: 100 µm. (L), Sequential interpolation between the mesh nodes with the deformation set returns a continuous tracking of the HT tissue (illustration related to e02). The points are random taken on the surface; they move in space following the tissue morphogenesis. The colour map is related to the position of the spots every hour. Yellow represents the position of the nodes at the initial time, orange after 1h, pink after 2h and purple after 3h. The grey line keeps track of the node trajectory. (M), Embryo during gastrulation. The tracking follows 25 cells for approximately 170 min. The arrows in green indicate the direction of cell motion, while the blue line tracks the entire path. (b), On the top, evaluation of the continuous error in 25 cells tracked during gastrulation at each 20 min interval. The dashed line indicates the slope (mG = 3.84) of the line that fits the median error values at each interval. Scale bar: 15 µm; video time resolution: 10 min; duration of the video: 170 min. (M'), On the bottom, error related to cell tracking during HT morphogenesis at each 20 min interval. The results are relative to tracking 30 cells during 216 min. The line fitting the median values of the intervals has a slope equal to mHT = 0.84. (M"), Evaluation of the tracking error normalised for the respective cell displacements. In blue, mean values and standard deviation are shown for mesodermal cell tracking during gastrulation at 20 min intervals. In grey, values related to HT cells.

Implementing the bidirectional registration, with the starting point fixed at N/2, ensured that the maximum cumulative error remained below one cell diameter across all videos.

## Optimization strategy

While high-resolution images provide maximum detail, non-rigid registration is computationally expensive. To identify an optimal balance between registration accuracy and computational cost, we evaluated the performance of the MIRT algorithm on progressively downscaled versions of the isotropic images.

Specifically, to evaluate the effect of image resolution on calculation time and registration accuracy, we created four test datasets by reducing the resolution to 10%, 15%, 20%, and 25% of the original isotropic images, using cubic interpolation (imresize3 in MATLAB).

For example, an isotropic image with a voxel size of 0.593 × 0.593 × 0.593 µm was downscaled to create lower-resolution versions with larger effective voxel sizes, ranging from approximately 5.93 × 5.93 × 5.93 µm (at 10% resolution) to 2.37 × 2.37 × 2.37 µm (at 25% resolution). The detailed methodology and full results of this optimization are described in the MATERIALS AND METHODS - Optimization Strategy section and in S1 Fig.

Fig 2F illustrates the results of this analysis, plotting computation time and tracking error against these four increasing levels of image resolution (from 10% to 25%). As expected, the computation time (black line) increases with image resolution. Conversely, the point registration accuracy (red line, tracking error) remains high and is minimally impacted by these changes in resolution.

Our goal was to determine the lowest resolution that would yield acceptable results without unnecessarily increasing the computational burden. We found that further increasing the resolution beyond 25% offered minor returns on accuracy at the cost of significantly longer computation times. To define an acceptable error threshold, we evaluated the percentage of landmarks with misregistrations exceeding the average cardiomyocyte diameter (20 μm) (Fig 2F'). Downscaling images to 25% of their original resolution ensured a misregistration rate below 1% for our longest time-lapse datasets (~6 hours). Given that the average acquisition duration was approximately 4 hours (S1 Table and S2 Fig), we selected the 25% resolution level as the optimal configuration for all subsequent motion estimation analyses.

## Continuous description of HT morphogenesis

The dense transformation fields, {Ti}, capture the motion of the entire imaged volume, including background and non-cardiac tissues. To focus specifically on myocardial morphogenesis, we first segmented the tissue slice by slice using three orthogonal planes in ITK-SNAP software (version 3.8.0) [14]. As the initial registration time, we manually segmented the image at t(N/2) (see MATERIALS AND METHODS – Image Segmentation), resulting in a binary z-stack of myocardial tissue (red mask in Fig 2I). From this binary mask, we generated a mesh using the vol2mesh function of the Iso2Mesh toolbox [15] (see MATERIALS AND METHODS – Mesh Generation). This mesh represents the myocardial surface, discretized into nodes of variable density between specimens. The mesh was refined using a Laplacian filter (*smoothsurf* in Iso2mesh) to reduce surface irregularities [16], and the resulting myocardial shape was designated as "'Live-Shape'". We then applied the *morphSurf3D* function sequentially to propagate each mesh node at t(N/2) forward or backward according to the transformation set {Ti}, thus generating a continuous motion profile of the myocardial surface. Iterating this process yielded a tracking of the heart tube surface (Fig 2L and S1 Video).

This procedure produced one 'Live-Shape' sequence for each live image dataset. Importantly, the method relies on the assumption of spatiotemporal continuity of the tissue. Therefore, before applying it to comparative morphogenetic analyses, we validated its robustness and tested its limitations, specifically assessing the impact of mitosis and cell mixing on the 'Live-Shape' mesh.

## Effects of cell division on 'Live-Shape' mesh

Validation of the sequential registration process demonstrated our ability to create motion profiles with cell-scale accuracy (S1 Fig). However, during the validation process, we had to exclude cells undergoing mitosis, since the algorithm assumes spatiotemporal continuity and does not explicitly account for division events.

To assess how cell division affects myocardial tissue continuity, we analyzed local tissue deformation and how it matches the deformation of the 'Live-Shape' mesh. Specifically, we manually segmented 8 cells before and after mitosis within a 2-h registration interval. The segmented cells at t–2h, t0h, and t+2h were converted into volumetric meshes. At t0h, the set of transformation fields was applied to the mesh nodes, and a line was fitted to the deformed nodes to extract the main deformation direction (Fig 2H, green and blue lines at t–2h and t+2h, respectively).

We then compared this deformation axis with the actual division axis, calculating the cosine similarity between the two. The results showed strong agreement: an average similarity of 0.87 (SD = 0.11) for cells at +2h and 0.87 (SD = 0.10) for cells at –2h (Fig 2H'). These findings indicate that the image-based algorithm not only captures the geometric changes caused by mitosis but also reproduces the anisotropic deformation in the correct direction, thereby supporting the robustness and accuracy of the method.

## Limitations of the approach

The continuous description employed did not allow the exchange of mesh node positions. Consequently, the observed registration accuracy at the cellular level suggests that during heart morphogenesis, cells largely maintain their relative positions within the timescales of our analyses. To test this idea, we evaluated the performance of our procedure on a

tissue known for high cell mixing: the nascent mesoderm during gastrulation. We applied the registration approach to 25 mesenchymal cells, which at this stage exhibit strong mixing behaviour and a chaotic pattern of movement ([17]; Fig 2M). A comparison of the tracking error over 170 min revealed a stark difference between the tissues (Fig 2M'). While the algorithm was accurate for gastrulating cells in the initial 20 minutes (error < 20 µm), the accumulated error subsequently increased approximately five times faster for mesenchymal cells than for cardiomyocytes (mG/mHT, where mG = 3.84 and mHT = 0.84).

This comparison highlights the algorithm's limitations with uncoordinated cell movements and, in turn, emphasizes the highly coordinated dynamics and low mixing behaviour of cardiomyocytes. Whereas mesodermal cells exhibited chaotic mixing over shorter distances, myocardial cells moved more cohesively and along straighter trajectories. This was reflected in the estimated tracking error, which, when normalized by cell displacement, was significantly higher for the mesoderm (Fig 2M"). This result not only defines the method's boundaries for fluid-like tissues but, more importantly, reinforces a key biological finding: the highly coordinated and solid-like behaviour of cardiomyocytes during heart morphogenesis.

Having validated our method at the single-embryo level, we then addressed the broader challenge of biological variability. Because developmental timing and morphology differ across embryos, individual 'Live-Shape' sequences cannot be directly compared or averaged. To enable cross-specimen quantification, we therefore integrated individual motion profiles into a consensus spatiotemporal reference by staging them against the Atlas.

### Integrating multiple live images into a consensus temporal reference

To compare different specimens and capture their intrinsic variability, it is essential to fix a spatiotemporal reference (see S2 Video). To address this challenge, our strategy involved two key steps: first, we developed a staging system that aligns live images temporally based on a morphometric feature. Second, we designed a strategy to overcome shape variability between equally staged hearts by projecting all specimens onto a unique spatial reference. This systematic approach for synchronizing live images in a common framework was achieved using the Atlas proposed by Esteban et al. as a temporal reference [9]. The Atlas offers a comprehensive and detailed description of the morphology of the heart tube at the tissue level during its development (E7.75-E8.5), from the cardiac crescent to heart tube looping. Esteban et al. discretized this developmental window into 10 groups using a morphometric parameter, d1/d2, to cluster the shape of 50 specimens (Fig 3A). This parameter is the ratio between the length of the boundary between the myocardium and the juxta-cardiac field (d1) and the length between the myocardium and the splanchnic mesoderm (d2) (as depicted by the green line in Fig 3A).

The lengths, d1 and d2, were not measurable on 'Live-Shape's due to missing parts; specifically, the IFTs and OFT as seen in Fig 2G (indicated by the yellow and blue arrow). To overcome this challenge, we identified alternative features that would produce similar staging results to d1/d2. We introduced three morphometric features, h/w, h/s, and θ (Fig 3B), which were computed by placing 7 specific landmarks on the 'Live-Shape' (MATERIALS AND METHODS - Selecting Myocardial Landmarks). The first two features, h/w and h/s were defined as the ratios between the Eulerian distances between the landmarks, while the third feature, θ, was calculated as an angle value. We excluded Atlas group 10 from our staging system as our live images collection did not cover the heart looping stage. Therefore, we calibrated our new staging system using 48 of the 50 samples from the Esteban et al. collection (E7.75-E8.25 12h). To define the three features, we manually selected the 7 landmarks (pt1-pt7) on the mesh of these specimens. To minimize the manual error, we selected the 7 landmarks three times and used median landmark coordinate values.

Once the 7 landmarks on the 48 shapes had been selected, we calculated the Euclidean distances between $pt_1$ and $pt_2$ (w) and between $pt_3$ and $pt_4$ (h). Furthermore, we defined θ as the angle formed between $pt_7$ and the points $pt_5$ and $pt_6$. These measurements showed a monotonic trend over time, which allowed us to correlate their value with the developmental stage (S3 Fig). To normalize the data and control for variations in myocardial size between specimens, we defined the morphometric features by taking ratios between these measurements, except for θ, which needed no correction for myocardial size.

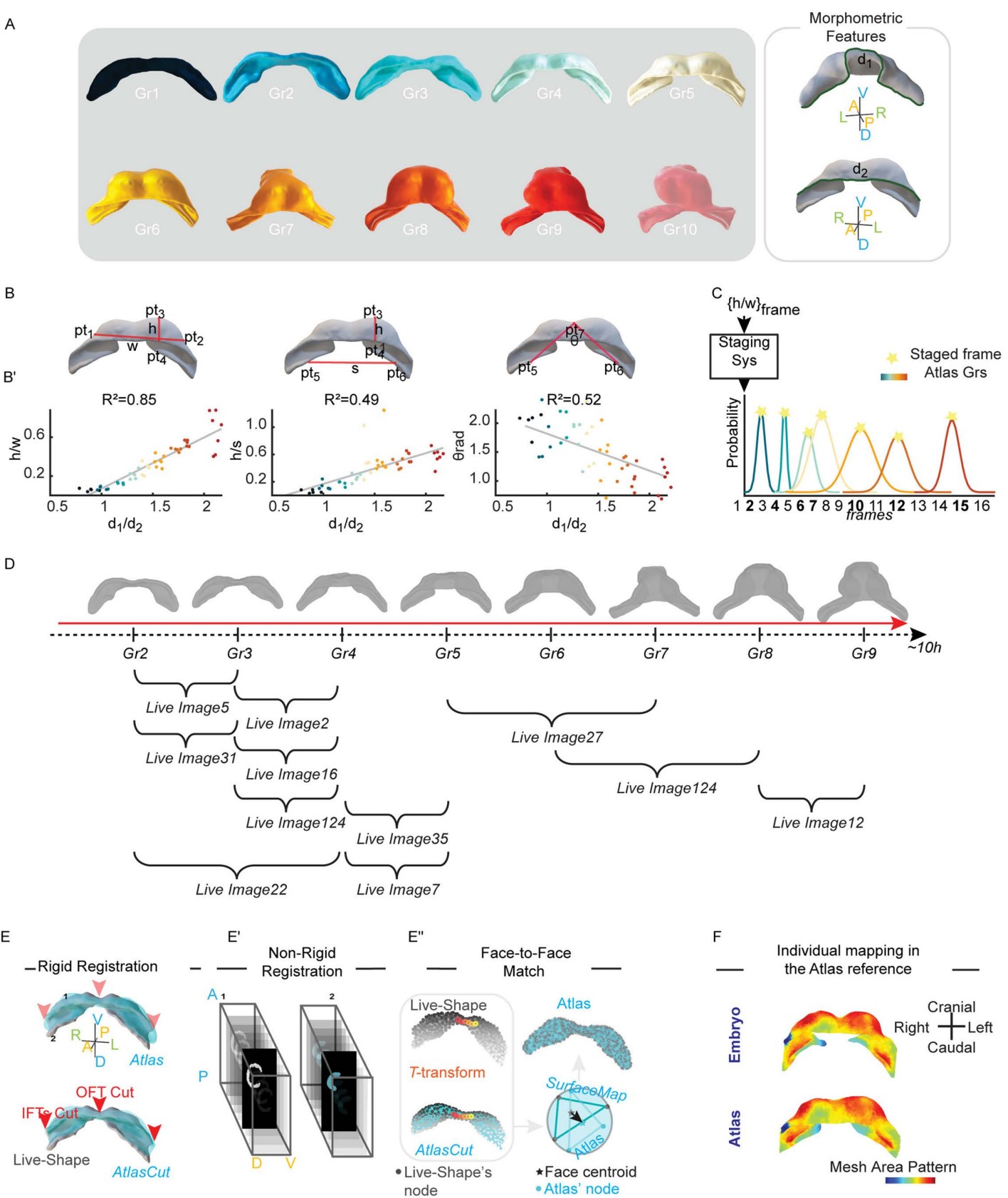

PLOS Computational Biology

**Fig 3. Integrating multiple live images into a consensus spatiotemporal reference by aligning each continuous motion profile with a published 3D+t Atlas.** (A), The static Atlas is described by 10 representative heart shapes. On the right, the morphometric parameter d1/d2 is shown. (B), Proposed features are defined as the Eulerian distances h, w, s, and θ. (B'), From left to right, a linear regression profile is shown for the ratios h/w, h/s, and θ versus d1/d2. Each scatter plot displays all Atlas specimens belonging to groups 1 to 9 (48 out of the total 50 specimens). The different groups are represented by a colour code. For each scatter plot, the coefficient of determination ($R^2$) is reported. The h/w ratio has the highest correlation value with with d1/d2, with $R^2$=0.85, while the linear regression models for h/s and θ returned $R^2$ values of 0.49 and 0.52, respectively. (C), Staging System: A Gaussian Mixture Model (GMMs) associates to the Atlas groups (indicated by colour code) the live image frame with highest probability (denoted by stars). (D), Results of the staging system. (E), TMM rigid registration rotates, translates, and resizes the Atlas shape (blue surface) until it overlaps with the 'Live-Shape' (grey surface). Parts of the Atlas shape corresponding to the missing IFTs and OFT in the 'Live-Shape' (indicated by red arrows) are removed, resulting in the Atlascut. (E'), Image non-registration (MIRT) is used to transform the 'Live-Shape' mask into the Atlascut mask. (E''), The T-transformation adjusts the position of the 'Live-Shape' nodes (grey point-cloud) to fit the Atlascut (blue point-cloud) morphology. The morphed shape is called SurfaceMap. The coloured points indicate the same nodes in their configuration before and after the transformation. Face-to-face matching between the Atlas shape and the 'Live-Shape' is performed between the centroids of the faces (marked by stars). (F), Validation of the spatial mapping is conducted by computing the mesh area in the 'Live-Shape' and plotting the growth values onto the Atlas face-to-face.

Subsequently, we evaluated which of the proposed features had the strongest correlation with $d_1/d_2$ by performing a stepwise regression (*stepwiselm* function in MATLAB) [18]. The results showed that the h/w ratio exhibited a high degree of correlation with the d1/d2 parameter, with a coefficient of determination ($R^2$) of 0.85 and a p-value of 2.30e-20 (Fig 3B'). Given the strong correlation of h/w with d1/d2 ($R^2 \approx 0.85$), we used this feature to stage our live images against the Atlas reference.

We then modelled the staging system as a Gaussian Mixture Model (GMM), assuming the h/w values are normally distributed for each Atlas group (MATERIALS AND METHODS - Modelling the Staging System; see also S4 Table). Then, we calculated h/w distances for each 'Live-Shape' extracted from the 3D+t images. The landmarks pt1, pt2, pt3, pt4 were fixed only on the 'Live-Shape' relative to the frame (N/2). The landmark positions for the remaining time-points were determined based on the continuous motion profile. This ensured that we were able to optimize time, but also to mitigate any potential bias that may have been introduced by manually selecting the landmarks (S3 Fig).

The classifier returned for each image frame the probabilities of belonging to the Atlas groups. The value closest to 1 determined the time-matching (Fig 3C). Since the temporal resolution of the live imaging is higher than the pseudo-temporal resolution of the Atlas, several time-lapse frames were often associated with the same Atlas reference (Fig 3D). To handle this issue, we selected only the frame with the highest probability among the equally clustered frames as the representative frame (S6 Table).

### Registering individual shape into the atlas reference geometry

Once the 'Live-Shape' sequences were temporally synchronized via staging, the next challenge was to reconcile their spatial variability. To achieve this, we used the Atlas not just as a timeline, but as a common spatial reference onto which each embryo's geometry could be mapped. This projection involved a three-step pipeline: 1. Rigid registration of Atlas shape with the 'Live-Shape'; 2. Non-rigid registration between 'Live-Shape' mask and the Atlas mask; 3. Face-to-face matching between the registered 'Live-Shape' and the corresponding Atlas shape.

The process began with a rigid registration of the Atlas to the 'Live-Shape' (Fig 3E; see also S5 Table). We employed a TMM-based algorithm which leverages the robustness of Student's t distributions against outliers [19,20](see MATERIALS AND METHODS - TMM setting parameters). This approach enhanced the overlap results, even in the presence of missing regions in the 'Live-Shape'. Once the two heart shapes were rigidly superimposed, we imported them into MeshLab and eliminated from the Atlas shape all nodes for which there were no correspondence with the 'Live-Shape', resulting into the 'AtlasCut' version (Fig 3E; MATERIALS AND METHODS - Removing Missed Correspondences).

The second step was aimed at identifying the necessary transformation to morph the 'Live-Shape' into the 'Atlas-Cut'. This was accomplished by performing a non-rigid registration in the image domain. We implemented the MIRT algorithm between the 'Live-Shape' mask and the 'AtlasCut' mask (Fig 3E'; MATERIALS AND METHODS - Defining

the 'Live-Shape' Mask, Defining the Atlas Mask). The resulting T-transformation was applied to the 'Live-Shape', modifying its nodes arrangement according to the 'AtlasCut' surface. This modified version of 'Live-Shape' is called SurfaceMap.

Crucially, this process standardizes the static geometry but preserves the unique dynamics of each embryo. We generate a SurfaceMap for each of a specimen's staged time points. The sequence of these SurfaceMaps over consecutive Atlas stages serves as a representation of that embryo's authentic motion profile, now displayed within the standardized coordinate system of the Atlas. The relative change in shape and position from one SurfaceMap to the next directly reflects the underlying dynamics captured in the original 'Live-Shape' sequence.

The final step of the spatial mapping resolved node density variability by establishing a definitive anatomical correspondence. We associated the centroid of each Atlas face to the nearest SurfaceMap face centroid (using the *knnsearch* function in MATLAB), resulting in a face-to-face matching between the two shapes (Fig 3E"). Validating the accuracy of this complex spatial mapping is inherently challenging, as no analytical ground-truth exists. We therefore assessed its efficacy by confirming that the anatomical context was preserved. As a proxy for anatomical consistency, we calculated the local area of each triangular face on the original 'Live-Shape' mesh (using the *meshFaceAreas* function in the geom3D library) and plotted these values onto the Atlas where correspondences were identified. We observed that the colour patterning was consistently preserved between the "Live-Shape" and the Atlas (Fig 3F), providing confidence that the trajectories and deformation patterns were being mapped to the correct anatomical region.

This entire process standardizes static geometry while preserving the unique dynamics of each embryo. For each specimen, we generate a sequence of SurfaceMaps corresponding to the staged Atlas time points. This sequence serves as a representation of the embryo's authentic motion profile, now re-plotted within the common coordinate system of the Atlas. As a result, the relative change from one SurfaceMap to the next directly mirrors the underlying dynamics of the original 'Live-Shape'.

With this spatiotemporal registration complete, we established a unified framework that overcomes biological variability. This common reference is the essential prerequisite for a comprehensive tissue deformation analysis, enabling two distinct quantifications: a stepwise analysis of deformation between consecutive Atlas stages, and a cumulative analysis that reconstructs the full deformation history.

**Extracting stepwise tissue deformation**

A crucial aspect of this quantification is that the deformation was always computed before the spatial projection, directly on the original "Live-Shape" geometry of each embryo. This sequence is critical because the mapping process itself adjusts the "Live-Shape" mesh to fit the Atlas; therefore, computing deformation *after* projection would measure registration artifacts, not the authentic tissue deformation of the specimen.

For each of the 12 qualifying embryos (see S6 Table), we first selected the specific 'Live-Shape' that best represented its classification into an Atlas Group. Gr1 was excluded from this deformation analysis, as our focus was on quantifying tissue changes directly related to morphogenesis events, whereas the transition from Gr1 to Gr2 primarily involves differentiation. The deformation was then computed between the selected native shapes by applying the principles of continuum mechanics between the rest state (the earlier stage) and the deformed state (the later stage) ([21]; MATERIALS AND METHODS - Principles of Finite Deformation Continuum Mechanics).

This analysis yielded two key kinematic descriptors for each triangular face of the mesh. The first, tissue growth rate (J), measures the local change in area. A value of $J > 1$ indicates that the tissue is actively expanding in that region, which could be driven by cell proliferation, cell growth, or changes in cell packing. Conversely, $J < 1$ signifies tissue compression. The second descriptor, tissue anisotropy ($\theta$), is the ratio of the principal eigenvalues of the strain tensor. This value quantifies how directional the deformation is. An isotropic deformation ($\theta \approx 1$) means the tissue is expanding or contracting uniformly in all directions, like an expanding balloon. An anisotropic deformation ($\theta > 1$) indicates that the tissue is being

stretched or sheared preferentially along a specific axis, a hallmark of processes like convergent extension or directed tube elongation. The principal eigenvector of the strain tensor reveals the orientation of this main axis of deformation.

This process resulted in maps of tissue growth rate (J) and anisotropy (θ) for each embryo, with the deformation pattern always plotted onto the shape of the end frame (Fig 4A).

Finally, only after this per-embryo calculation was complete, were the resulting deformation maps projected onto the common Atlas geometry for comparison. This final mapping step used the Atlas as a common canvas to display the individual deformation data. Using the established anatomical correspondences, the deformation values (J and θ) from each "Live-Shape" face were transferred to the corresponding face on the Atlas shape (Fig 4A'). The outcome is a set of stepwise deformation maps where the underlying geometry is standardized, but the colour pattern on the surface reflects the authentic, individual deformation pattern of each specific embryo. This collection of standardized datasets enabled us to investigate the spatial consistency of these patterns by calculating their mean (μdef) and standard deviation (σdef) for each developmental step.

## Calculating cumulative deformation

The ultimate goal of our framework is to reconstruct the cumulative deformation that shapes the heart over the entire developmental timeline (Gr2 to Gr9). This requires concatenating the stepwise deformation information from our fragmented, individual datasets. However, two major challenges prevent a simple concatenation. First, the static Atlas itself lacks any kinetic information that links one stage to the next. Second, while the SurfaceMaps project each embryo onto the Atlas shape, they each retain a unique mesh structure (point density), making a direct, point-by-point averaging across specimens impossible.

To overcome both challenges, we developed a two-step strategy. First, we established a common set of anatomical reference points at which to evaluate deformation. Second, we computed the mean stepwise deformation at each of these points for every developmental transition.

To define this set of reference points, we selected the dense point distribution of a single, centrally-staged SurfaceMap (from specimen e27) to serve as our reference 'eye-PC' (eye point cloud). The nodes of this mesh became our set of fixed anatomical loci for the entire analysis. We then used a 'chain-of-correspondences' approach to find the corresponding position of each of these loci in all other Atlas stages. This was done by linking the 'eye-PC' to the SurfaceMaps of other embryos (the 'hook-PCs') that covered the adjacent time intervals, using nearest-neighbour correspondences (Fig 4B and 4C; see MATERIALS AND METHODS - Concatenating Individual HT Motion Profiles to Compute Cumulative Deformation). This process defined a consistent coordinate system, a set of corresponding anatomical points, present across all stages from Gr2 to Gr9.

With this common set of reference points established, we could robustly compute the mean deformation. For each developmental step (e.g., Gr2→Gr3) and for each reference point $i$, we gathered the deformation values (J and θ) from all individual embryos covering that transition and calculated the mean stepwise deformation ($\overline{J_{gr}}$, $\overline{\theta_{gr}}$).

Finally, the long-term cumulative deformation was computed as the product of these sequential mean deformation values. For each i-correspondences, the cumulative growth ($\overline{J}$) and anisotropy ($\overline{\theta}$) were calculated as:

$$\overline{J} = \prod_{gr=2}^{9} \overline{J(i)_{gr}},$$

(1)

where $\overline{J(i)_{gr}}$ is the average growth for a single step between consecutive stages.

$$\overline{\theta} = \prod_{gr=2}^{9} \overline{\theta(i)_{gr}},$$

(2)

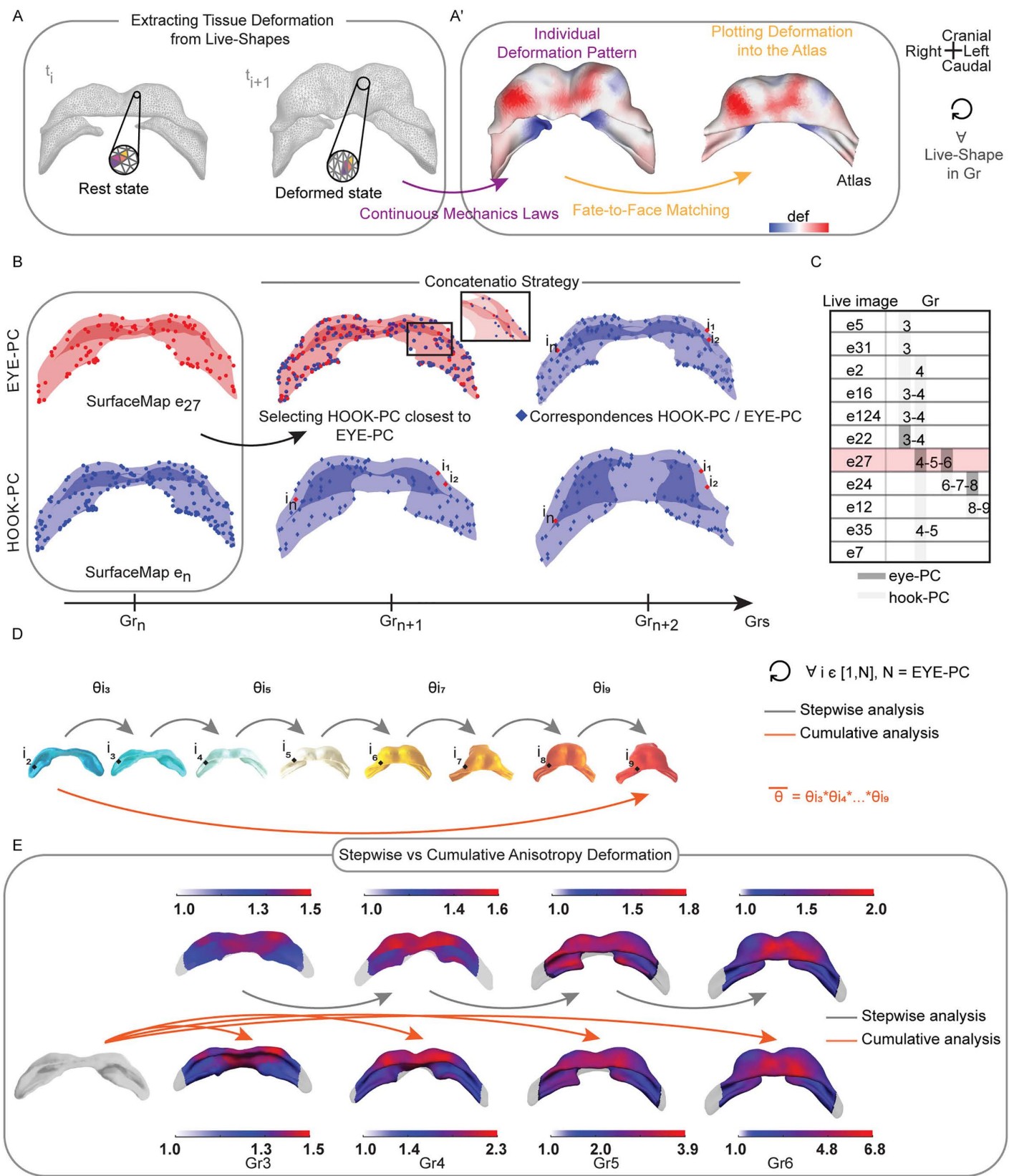

**Fig 4. Quantifying tissue deformation during early morphogenesis.** (A), The triangles of the mesh were transformed according to the T-transformation from ti to ti+1. Colours code indicated the same triangles in their rest and deformed state (shapes refer to e27). (A'), The tissue deformation pattern into is plotted in 'Live-Shape'(ti+1) and mapped face-to-face into the synchronise Atlas group. We compute the individual deformation pattern for each staged 'Live-Shape' and for each Gr. (B), Schematic overview of the pipeline for concatenating multiple motion profiles using the EYE-PC and HOOK-PC strategies. SurfaceMap e27 at Grn represents the EYE-PC (red shape and dots). To predict the position of SurfaceMap e27 in subsequent groups (Grn+1, Grn+2), we used known SurfaceMap from another specimen, referred to as the Hook-PC (blue shape and spots). The closest points in the HOOK-PC to the EYE-PC at Grn were selected as corresponding points (blue rhombuses). The position of SurfaceMap e27 (i1, i2,..., in) in Grn+1 and Grn+2 was then determined by the positions of the selected points from the Hook-PC. (C), Live images concatenation path. Embryo e27 is the reference. In dark grey the eye-PCs are highlighted, in light grey the hook-PCs. (D), The cumulative anisotropy rate ($\theta\_$)of each mesh face (i) between two arbitrary Grs is given by the product of all the intermediate $\theta\_gr$. (E), Stepwise vs Cumulative anisotropy ($\theta$). Cumulative deformation is computed from Gr2. Colour bars indicate the deformation magnitude. $\theta = 1$ isotropic deformation. $\theta\_ > 1$anisotropic state. Caudal view of the heart tube is shown.

where $\overline{\theta(i)_{gr}}$ is the average anisotropic stretch for a single step.

The resulting maps (Fig 4E) are the final output of our method, providing a quantitative picture of the tissue mechanics that shape the heart. For the first time, these maps reveal a clear spatial separation of these mechanics: tissue growth is concentrated on the sides, while the middle is stretched in a distinct direction. These two actions work together to transform the flat sheet of early heart cells into a three-dimensional tube.

### In-silico fate map

The same concatenation principle allowed us to develop our final and most powerful tool: an in-silico fate map to track cardiomyocyte trajectories through morphogenesis (see MATERIALS AND METHODS - Concatenating Individual HT Motion Profiles to Compute in Silico Fate Map). While the cumulative deformation analysis relies on averaging motion profiles to capture a consensus behaviour, the goal of a fate map is different: to create a single, continuous, and anatomically realistic trajectory for a 'pseudo-cell'.

Averaging the motion from different embryos, particularly those with inconsistent missing parts in the IFT and OFT could lead to a physically unrealistic, discontinuous motion profile. To prioritize anatomical coherence and create a smooth trajectory, we therefore adapted our concatenation strategy. Instead of averaging, for each transition between Atlas groups, we selected the single, most complete 'Live-Shape' motion profile from our collection to serve as the 'link' in our concatenation chain.

By carefully selecting a path of the most complete individual specimens (e.g., e31, e16, e35, e27, e24, e12, as illustrated in Figs 5A and S4), we constructed a continuous and anatomically coherent motion profile for a single reference mesh spanning from Gr2 to Gr9. The result of this process is a 'Dynamic Atlas': a high-fidelity descriptive model that maintains the canonical morphology of the static Atlas at each stage [9], but with mesh nodes that now follow a realistic trajectory derived from the concatenated live imaging data (Fig 5B and S3 Video).

### Validating the fate map deformation

To evaluate how well the Dynamic Atlas predicts myocardial deformation, we validated its ability to reproduce the myocardial growth patterns observed in live specimens. To do this, we compared the average growth profiles of 10 selected zones in our live imaging data with those predicted by the Dynamic Atlas. These zones were identified in the Dynamic Atlas at Gr9 (as shown in Fig 5C) and tracked backwards across other groups (Fig 5C').

To determine the Dynamic Atlas growth profile, we calculated and smoothed the mesh areas for each zones using the *meshFaceAreas* function in the MatGeom toolbox. For each zone, we summed the area values, defining the Atlas growth profile, shown in Fig 5E' as three-degree B-splines. To determine the 'Live-Shape's growth profile we identified the same 10 zones as illustrated in Fig 5D This involved first rescaling each specimen to the reference static Atlas shape using the Coherent Point Drift (CPD) algorithm [12] (step 1 in Fig 5D). We then computed and smoothed the mesh areas for

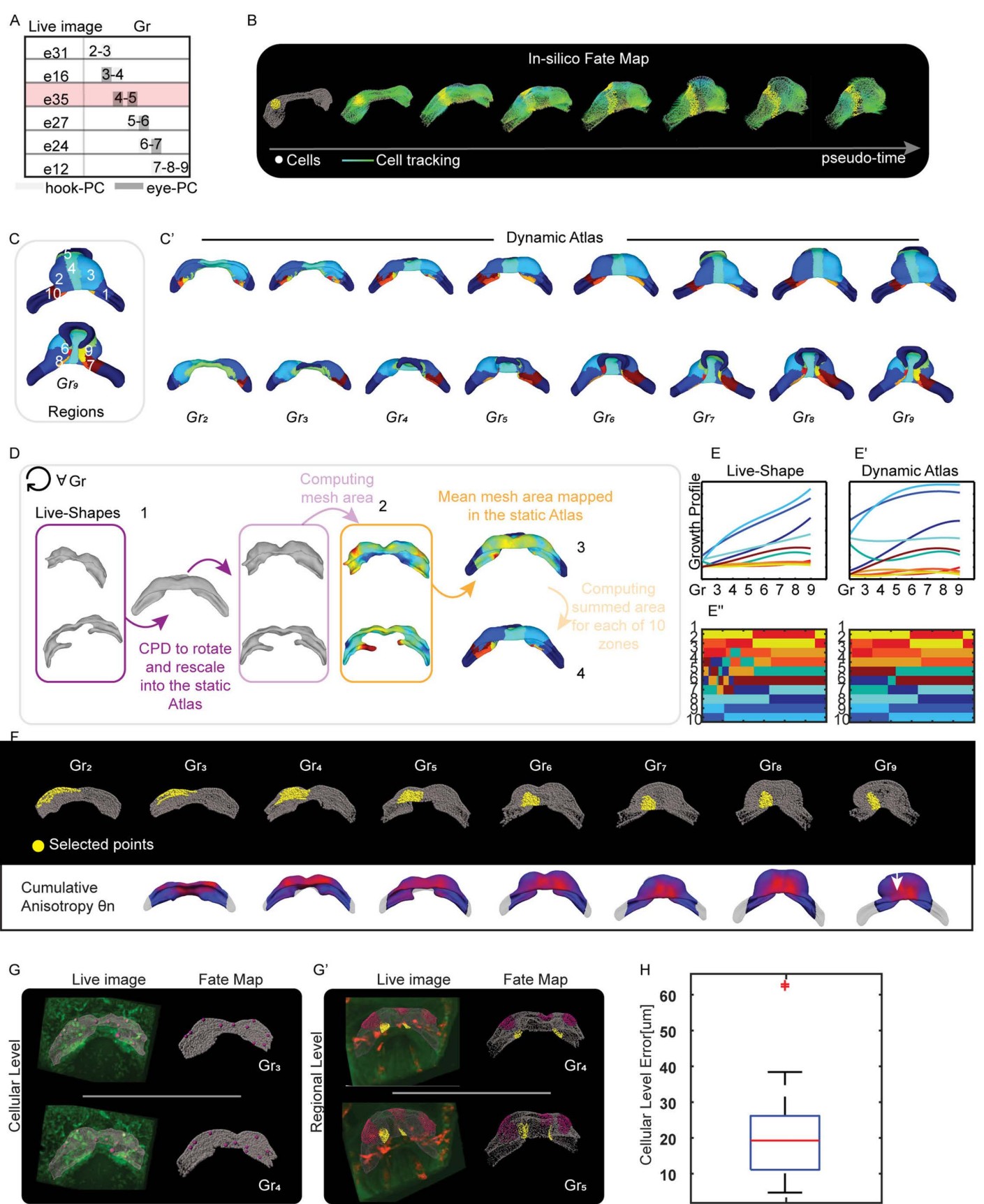

**Fig 5. Creating an in-silico fate map to analyse heart tube morphogenesis at both cellular and regional levels.** (A), Concatenation path of the individual live image. We concatenated through the hookPC and eye-PC strategy the videos related to embryos e31, e16, e35, e27, e24 and e12. The reference embryo is e35. (B), In-silico fate map tracking early cardiac morphogenesis as a continuum. The tool in Imaris allows to select points or regions in the myocardium and follow their motion over time from Gr2 to Gr9 in Atlas. In grey the point-cloud, in yellow the selected points track the regional changes at different stages. The lines, on the other hand, draw the displacement of the point-cloud. (C), The selected 10 arbitrary zones of the myocardium on the Gr9 shape. (C'), The displacement of the 10 zones in the different groups of the Dynamic Atlas. The two rows above represent the anterior part of the heart, the two rows below the posterior part. (D), Strategy to identify the same 10 anatomical zones in the live embryo. For each group, 'Live-Shape's were rescaled with a rigid registration algorithm to the Corresponding Atlas shape. Then the area of each mesh was calculated (colour map) for each rescaled 'Live-Shape's. Finally, the area values were averaged within the Atlas reference and summed for each of the 10 areas. (E-E'), The 'Live-Shape' and the Dynamic Atlas growth profiles of the 10 zones, interpolated for each Grs with a 3degree B-Spline. (E") The validation strategy. (F) Top: A selected region in the Dynamic Atlas, illustrating how the region deformed anisotropically over time. Bottom: Cumulative anisotropy from Gr3 to Gr9. (G), At the cellular level (left), we tracked cell positions in e02 at Gr3 and Gr4 using Imaris (pink spots). On the right, we marked the initial positions of these cells on the fate map at Gr3 and then evaluated their predicted final positions at Gr4 (pink spots). (G'), At the regional level, we tracked different groups of points on the myocardium mesh of e27 from Gr4 to Gr5 (pink and yellow points). On the right, we identified the same groups of points on the fate map at Gr4 and assessed where these points ended up in Gr5. (H), Cellular-level error across 40 tracked cells from e01, e02, e05, and e16, compared to the corresponding predicted positions from the dynamic atlas. On average, the error is 20 μm, roughly equivalent to a cell diameter. Only 2 cells show an error greater than twice the cell diameter. We consider 95% of the cells to be well-predicted.

these rescaled 'Live-Shape's (step 2 in Fig 5D). Next, we mapped and averaged the individual mesh areas onto the static Atlas (step 3 in Fig 5D). As a last step, we found the 10 zones as the correspondences between the static Atlas and the Dynamic Atlas (step 4 in Fig 5D). The 'Live-Shape' growth profile is shown in Fig 5E'.

We compared the growth profiles by checking their consistency at each x-point (Fig 5E"). We verified this consistency by correlating the sorted growth values of each zone between the two profiles across the groups (Fig 5E'). This method allowed us to examine myocardial growth profiles across different heart zones, independent of growth magnitude, while accounting for embryonic variability. Our analysis found that the growth profile of the Dynamic Atlas matched that of the 'Live-Shape's with 81% accuracy (see also S5 Fig). The error rates for zones 1, 10, and 5, corresponding to the left and right IFT and OFT, respectively, were relatively high, likely due to manual cutting and misalignment issues during spatial mapping. Removing these zones improved the validation accuracy to 92%. At the regional level, we also evaluated tissue deformation, finding that the Dynamic Atlas also qualitatively captured key deformations. For example, the ventral part of the growing ventricle underwent deformation in the craniocaudal direction (Fig 4E), and the fate map deformed anisotropically in the medial area, as shown in Fig 5F.

### Validating the fate map motion

We then validated the fate map's predictive power at the cellular level. We performed virtual labelling of single cells or regions within the Dynamic Atlas and compared their predicted fate with the ground-truth trajectories tracked directly in the live images (Fig 5G–G').

To quantify this accuracy, we tracked the real-world positions of 40 cells from their initial to final locations in the live images. Subsequently, we compared these final positions with those predicted by the fate map. Given the intrinsic shape variability, we validated the predicted positions manually: a prediction was considered correct if it fell within a small target area (~2 cell diameters) surrounding the ground-truth final position. Using this criterion, we found that the fate map correctly predicted the location of 95% of the selected cells (S7 Table).

### Discussion

We designed and developed an image-based pipeline to extract and compare tissue motion and deformations during early morphogenesis, applying it here to mouse heart development. Our method offers a computational solution to biological challenges and overcomes technological limitations in mammalian organogenesis. This approach enables the extraction of myocardial motion and deformation while successfully managing the intrinsic variability of the phenomenon.

Our strategy contrasts with trajectory-based Lagrangian methods, which reconstruct deformation from the tracked paths of individual components. Prominent examples include the discrete analysis of Kawahira et al. [22] for C-looping in the chicken heart, where the authors quantified myocardial torsion from the tracking of several hundred cardiomyocytes, and broader frameworks that have been shown to predict the embryonic fate map from complete deformation trajectories in systems like the zebrafish embryo [8]. However, the application of such methods relies on access to continuous, long-term recordings of the entire morphogenetic process, a prerequisite rarely met in mammalian organogenesis.

Such a cell-tracking-based strategy would not be feasible for the drastic deformations that many tissues undergo during organogenesis, as is the case for heart tube formation. In fact, early cardiac morphogenesis involves complex shape changes and topology, and a detailed reconstruction of deformation maps would require tracking thousands of cells in a dense population. This challenge is compounded by the limited spatial and temporal resolution of mouse heart images and photobleaching. Our strategy was specifically developed to circumvent these difficulties by leveraging image intensity directly, making it suitable for the fragmented and incomplete datasets common in most developmental biology labs. The validation process showed that our registration method succeeds in tracking motion profiles with cell-scale accuracy (Figs 2F' and S2D). Furthermore, the method revealed the coordinated nature of cardiomyocyte motion, denoting low cell mixing during the shaping process (Fig 2M–M"), confirming the appropriateness of our continuum-based approach for this tissue.

To compare different specimens, we needed to address the intrinsic variability of HT morphogenesis. Combining and comparing multiple kinetic datasets required defining a common reference system. We chose the Atlas shapes as our spatio-temporal template [9]. The Atlas not only reduces the number of developmental stages to consider but also resolves issues arising from incomplete shapes. This Atlas-based integration is the key feature that allows our framework to handle fragmented data from multiple embryos, a challenge not typically addressed by classic Lagrangian analyses, which are ideal for single, complete datasets.

We introduced a standalone morphometric staging system that synchronizes live images along the Atlas's temporal axis, employing a simple geometric feature (w/h). This feature offers a significant advantage as it can be applied directly to 2D images, without requiring complex 3D reconstructions. In contrast, the staging system proposed by Esteban et al. [9] relies on a parameter (d1/d2) derived from geodesic distances that necessitates a full 3D reconstruction and complete heart capture, requirements difficult to meet in live imaging. Our statistical analysis showed a strong correlation between our w/h feature and the d1/d2 parameter (Fig 3B'), demonstrating that w/h is a robust and effective alternative for mouse heart staging.

To address shape variability within the same stage, we implemented spatial mapping through non-rigid image registration. This approach proved more efficient and robust than point cloud–based methods. By converting point clouds to binary image masks, we could match a significantly greater number of anatomical points and ensure accurate correspondences, as validated by the consistent preservation of local mesh area patterns between the 'Live-Shape' and the Atlas (Fig 3F).

Deformation patterns were derived from the motion profile of each specimen. Comparing these patterns across embryos required the standardized reference frame the Atlas provides. The innovation of our cumulative method lies in its ability to recreate a complete deformation profile by linking multiple, incomplete live images. Acquiring single live images spanning the full early heart developmental timeline is exceptionally challenging and demands high-performance microscopy systems. Therefore, our cumulative strategy provides a flexible and practical solution for computing deformations between any selected stages.

The quantitative framework presented here offers a powerful platform to investigate the biological drivers of morphogenesis. Our analysis reveals two fundamental principles of early heart morphogenesis: predominant local cellular coherence and a strong spatial compartmentalization of tissue mechanics. We observed that, while individual cells generally maintain their neighbours, the tissue undergoes dramatic and highly organized deformation. Specifically, regions of high

isotropic growth are distinct from those exhibiting pronounced anisotropic stretch. A detailed discussion of these biological implications is presented in a companion manuscript [23, under review].

By integrating motion profiles from multiple specimens, we created the first in-silico fate map for early heart tube morphogenesis, establishing a Dynamic Atlas. This digital model accurately reproduces regional growth patterns and tissue anisotropy. We validated the fate map's reliability by tracking 40 cell positions on live images, showing it effectively models cardiomyocyte locations across developmental stages.

Based on these results, we propose the Dynamic Atlas as a powerful exploratory tool. It enables researchers to select and track pseudo-cells, providing a novel approach to explore cardiac morphogenesis at both cellular and regional scales. This innovative tool illuminates regionalized cell coordination and raises new, testable hypotheses about the biological and genetic factors governing cardiac development.

### Limitations and future directions

The framework presented here provides a quantitative and descriptive account of average tissue mechanics and their variability during early heart morphogenesis. A limitation of the current study is the relatively small number of specimens available for each discrete Atlas stage, which precludes the construction of robust, inferential statistical models. The descriptive approach we have adopted is therefore the most scientifically rigorous for the present dataset.

However, our Dynamic Atlas is designed to be an expandable resource. As more live imaging data becomes available, this framework will serve as the foundation for future, more sophisticated statistical analyses. With a larger cohort, one could build probabilistic models to perform statistical inference, for example, to test hypotheses about how genetic mutations affect the probability distribution of tissue deformation. Furthermore, generative statistical models could be trained on this richer dataset to create in-silico simulations of development, allowing for the prediction of morphogenetic trajectories under different conditions. Our work thus provides both the foundational methodology and the initial dataset for a future, fully statistical atlas of mammalian organogenesis.

## Materials and methods

### Ethics statement

Animals were handled in accordance with CNIC Ethics Committee, Spanish laws and the EU Directive 2010/63/EU for the use of animals in research. All mouse experiments were approved by the CNIC and Universidad Autónoma de Madrid Committees for "Ética y Bienestar Animal" and the area of "Protección Animal" of the Community of Madrid with reference PROEX 220/15.

### Experimental workflow

**Mouse strains.** For this study, mice were maintained on mixed C57Bl/6 or CD1 background. We used the following mouse lines (S1 Table), which were genotyped by PCR following the original study protocols. Male and female mice of more than 8 weeks of age were used for mating.

**Embryo culture and live imaging of gastrulating mouse embryos.** Live imaging procedures followed the protocol outlined in [24]. In brief, mouse embryos were carefully collected and dissected within a dissection medium comprised of DMEM supplemented with 10% fetal bovine serum, 25 mM HEPES-NaOH (pH 7.2), and penicillin-streptomycin (50 µg/ml each). For embryos spanning E6.5 to E7.5, culture conditions were established using a mix of 50% Janvier Labs Rat Serum Sprague Dawley RjHan SD (male only) and 50% DMEM FluoroBrite (Thermo Fisher Scientific, A1896701) with incubation at 37°C and a 7% $CO_2$ concentration. Imaging was conducted on a Zeiss LSM780 platform, featuring a 20× objective lens (NA=1) and a MaiTai laser set at 980 nm for two-channel twophoton imaging. Fluorescence was detected with Non Descanned Detectors equipped with the filters cyan-yellow (BP450-500/BP520-560), green-red (BP500-520/

BP570-610) and yellow-red (BP520-560/BP645-710). Zen software (Zeiss) facilitated data acquisition with an output power of 250 mW, pixel dwell time of 14.8 s, line averaging of two, and an image dimension of 1024 × 1024 pixels.

## Computational workflow

In this section, we provide a detailed explanation of the steps within the framework. We outline the computational strategy employed to extract, measure, and compare deformation patterns from 3D time-lapses, by integrating image processing, computer vision, statistical analysis, and physics concepts.

**Image preprocessing.** The raw images were captured using a two-photon microscope (Zeiss LSM780) [5]. Some images were significantly affected by shift. To address this, we applied block matching registration to correct for both translational and rotational misalignments across all planes, using the registration tool developed by [25]. Most images included two reporter alleles, corresponding to two channels (ch1 and ch2). To integrate the information from both channels, we combined ch1 and ch2 using Fiji [26] to produce a single-channel image, I(x, y, z, t). This combined image was then imported into MATLAB via the JAVA MIJ package MIJ (*ij.IJ.openImage* function, *ImagePlus2array* function) [27,28] and cropped using the *imcrop3* function to eliminate non-cardiac tissue and reduce the image size.

In the raw data, voxels had varying dimensions along the x, y, and z axes (anisotropic voxel [aµm; aµm; bµm], see S1 Table – VoxelRes column). To ensure dimensional comparability, we resampled the images using nearest-neighbour interpolation. This process computationally adds new slices along the z-axis, matching the z-axis resolution to the in-plane resolution (x, y). This process converted the anisotropic voxels into isotropic voxels with uniform dimensions of [aµm; aµm; aµm] on all three axes. This resampling was performed with the *imresize3* function. To reduce noise, we applied a 3D Gaussian filter (*imgaussfilt3*) with a sigma value of 0.5 as needed. Next, we scaled the intensity range of each 3D image to [0–1] using the *mat2gray* function to mitigate the effects of photobleaching. Finally, we rotated the hyperstacks by 90° around the y-axis in the transverse plane using the reslice function in Fiji. The processed hyperstacks were then saved in the Tagged Image File Format (TIFF).

**MIRT algorithm.** In MIRT (Medical Image Registration Toolbox) non-rigid registration algorithm [12,29,30], the alignment between two images, I and J, is accomplished by minimizing a target function, $E_{target}$ (eq.3). This target function is composed of three key components: the similarity measure ($E_{sim}$), the transformation model T(x), and the optimization method:

$$E_{target}(u) = E_{sim}\left(I, J\left(T(x)\right)\right) + \lambda E_{ref}\left(u(x)\right)$$

(3)

As similarity function $E_{sim}$, we have decided for the Sum of Squared Differences (SSD).

where N is the total number of voxels. $E_{sim}^{SSD}$ quantifies the intensity difference for each voxel position in the 3D images domain Ω. SSD is one of the simplest intensity-based similarity measures. Nevertheless, it guaranteed good results when applied to live heart images. This was because consecutive images exhibited a high degree of similarity, owing to the temporal resolution being sufficiently high.

As transformation function T(x), the algorithm uses Free-Form Deformation (FFD) transformation [31], which generates smooth deformations. This transformation model parameterized deformations on a per-pixel basis, i.e., it deforms a 3D image (Ω = (x,y,z)| $0 \leq x_n \leq N$, $0 \leq y_m \leq M$, $0 \leq z_k \leq K$) by manipulating a regular grid of control points (blue grid showed in Estimating Individual Live Image Motion in Fig 1) that are distributed across the image at an arbitrary mesh resolution (nx x ny x nz). The spacing of the regular grid points governs the transformation scale. At each iteration, the control points update their position (orange grid showed in Estimating Individual Live Image Motion in Fig 1), and a dense deformation is computed by using a B-spline transformation [32].

We fix the mesh resolution (nx x ny x nz) at 10x10x10µm, which is equivalent to half the average diameter of cardiomyocytes (Fig 2G).

The regularization parameter, $E_{reg}$, was calibrated using an empirically determined value of $\lambda = 0.01$, and the tolerance and maximum number of iterations were set at $10^{-8}$ and 100, respectively, as stopping conditions. We used a four-level hierarchical approach as optimization method.

**Optimization strategy.** To evaluate the performance of the MIRT algorithm on the downscaled images, all parameters were kept consistent with those described previously, except for the grid resolution. This parameter was defined based on a constant biological scale: the grid spacing was set to approximate half the average diameter of a cardiomyocyte (~10 μm).

Since the physical size of each voxel changes depending on the downsampling level, the grid resolution (expressed in voxels) had to be adjusted accordingly for each test case to maintain this constant physical spacing. The specific relationship between the downsampling level, the resulting voxel size, and the corresponding grid resolution used is summarized below:

- For the 10% resolution images (voxel size ~5.93 μm), the grid resolution was set to 2 voxels.

- For the 15% and 20% levels (voxel sizes ~3.95 μm and ~2.97 μm), it was set to 3 voxels.

- For the 25% level (voxel size ~2.37 μm), it was set to 4 voxels.

**Image segmentation.** We accurately segmented the cardiac tissue at frame (N/2), with N equal to the total number of time points in the image. Segmentation was performed manually using the open-source software ITK-SNAP (version 3.8.0) [14] by the polygon inspector tool.

We used 3D I/O (IJ Plugins) in Fiji to convert the TIFF format in MetaImage Medical Format (.mha) format compatible with ITK-SNAP.

Segmentation was saved as The Neuroimaging Informatics Technology Initiative (.NIfTI) file (using the nifti_io.jar plugin from Fiji) and imported into MATLAB. In MATLAB, the *fillholes3d* function was applied to the segmented images to fill any hole present by setting the maximum gap to be equal to 1 voxel.

**Mesh generation.** To create a discretized version of heart tissue, we generated triangular (2- simplex element) meshes of each segmented heart using Iso2Mesh [15] a MATLAB/Octavebased mesh generation toolbox. Because myocardial tissue is sufficiently thin (30 μm or 1–2 cell layers), we created a surface mesh. We used a CGAL surface masher to create a genus-0 surface from the segmented image. The *vol2mesh* function was run with an isovalue fixed to 1, the maximum tetrahedral volume set to 4 voxels, and the maximum radius of the Delaunay sphere set to 4 voxels. Each surface was then smoothed with a Laplacian filter (*smoothsurf* function in Iso2Mesh), setting the smoothing parameter equal to 0.9. The algorithm was iterated 5 times. This step allowed us to have a smoother mesh surface and to eliminate possible inaccuracies associated with manual segmentation. The resulting mesh was then saved as Polygon File Format (.PLY) (*write_ply* function in toolbox_graph toolbox) and visualized in MeshLab (version 2020.06) [Cignoni et al., 2008], an open-source system for processing and editing 3D triangular meshes. In MeshLab, we run the Isotropic Explicit Remeshing function to improve aspect ratio (triangle quality) and topological regularity.

**Selecting myocardial landmarks.** We identified morphological points on the myocardial 'Live-Shape' as follows:

- $pt_1$-$pt_2$ were taken at the border between the ventricle and the right and left IFT respectively;

- $pt_3$ - $pt_4$ were selected at the maximum elongation of the ventricle;

- $pt_5$-$pt_6$ were taken at the border between the cardiac mesoderm and the splanchnic mesoderm at the bending point of the IFT related to the foregut invagination;

- pt7 is a specific point located on the ventricle of the heart. This point was determined by the intersection of the myocardium and the longitudinal axis of symmetry, which was determined using Principal component analysis (PCA) in the mesh nodes. In cases where the shape of the heart was strongly asymmetric, due to missing parts of the IFTs, the

point of asymmetry was manually corrected. We forced the shapes of the hearts to the centre of the axes, subtracting the mean values of the entire node's distribution from each node. PCA defines the main 3 axis of the distribution of the nodes, considering the area of the mesh. The reason for considering the mesh area was that the faces of the mesh do not have the same area, particularly in smaller anatomic areas. Without weighing, PCA could imbalance the orientation of the heart shape, leading to inaccuracies in the determination of $pt_7$.

**Modelling the staging system.** We modelled the staging system as a Gaussian Mixture Model (GMMs). We examined whether the h/w values for each class followed a normal distribution. Due to the small number of specimens in each Atlas group (as reported in S3 Table), we performed a Shapiro-Wilk test to determine normality with IBM SPSS Statistic [33]. The results showed no evidence of nonnormality, as shown in S4 Table. Therefore, after a visual examination of Normal Q-Q plot, we decided to model the Atlas groups as a convex combination of normal distributions $X_c$, parameterised by $\{\mu_c, \sigma_c^2\}$ with c = 1...9 groups.

We associated the frames with the highest probability to the Atlas group. For the first and last frames where the matching is uncertain, we decided to classify only those with a probability of affiliation greater than 75%. We chose to exclude e06 due to its early stage of development. The e26 was excluded from the deformation analysis due to significant missing parts (IFTs). The staging of e01 was initiated from frame 11, as initial frames indicate that the embryos exhibit obsolete movements associated with stabilization outside the uterus. Conversely, e05 staging began at frame 14, since prior frames correspond to the period when the cardiac crescent is forming. During this period, the main motion estimation errors arise due to the incorporation of secondary heart field cells.

**TMM setting parameters.** The number of mixture components was determined by dividing the number of 'Live-Shape' nodes in half. This number varied for each specimen and was determined by testing and adjusting to achieve a proper balance between computational efficiency and precision of registration. The EM algorithm was limited to a maximum of 150 iterations as a registration stop condition. The shapes were cantered on (0,0,0) before being registered, decreasing the computational cost.

**Removing missed correspondences.** After aligning the Atlas and 'Live-Shape' using rigid registration, the shapes were imported into MeshLab and Atlas nodes that lacked correspondences with the 'Live-Shape' were meticulously selected with the *Interactive Selection tool* and removed. The modified Atlas shape was referred to as 'AtlasCut'.

**Defining the 'Live-Shape' mask.** We applied the set of transformation fields $\{T_i\}$ to the segmentation of the myocardium relative to frame N/2 by a sequential B-Spline interpolations (with *mirt3D_transform* function in MIRT toolbox [29]). This approach returned a 3D + t binary image of the myocardium during its shaping. We selected only the frames staged into the Atlas.

**Defining the Atlas mask.** The Atlas binary mask was obtained by converting the 'AtlasCut' shape into a 3D image. To achieve this, the 'AtlasCut' was first restored to a genus-0 surface running MeshLab's *closeholes* function. If the mesh had improper meshes, the shell had discontinuities that were manually closed. The mesh was then transformed to a shell of voxel in a 3D image using the *surf2volz* function in the Iso2mesh toolbox [15] with grid resolution equal to 1 voxel.

The surface mask was then converted into a full binary mask by filling the gaps.

**Principles of finite deformation continuum mechanics.** To compute the deformation, we defined the rest triangular frame (R) and the deformed triangular frame (T) into the world space as:

$$R = \begin{bmatrix} e_1 x & e_2 x \\ e_1 y & e_2 y \end{bmatrix} \qquad\qquad T = \begin{bmatrix} f_1 x & f_2 x \\ f_1 y & f_2 y \end{bmatrix} \tag{4}$$

with $e_1 = r_2 - r_1$, $e_2 = r_3 - r_1$ with $r_n$ equal to the 3D position of the element nodes. $f_1 = m_2 - m_1$, $f_2 = m_3 - m_1$, with $m_n$ equal to the 3D position of the element nodes.

Then, we represented the deformation gradient F (eq.5) as the transformation from the rest triangular frame (R) into the deformed triangular frame (T):

$$F = TR^{-1} \tag{5}$$

In particular, the growth rate J of a mesh element was determined by the Jacobian determinant of F. This value indicated how the area of the mesh element had changed due to the deformation. If the area is preserved, the deformation is isochoric, and J = 1. On the other hand, J > 1 indicates tissue expansion, while J < 1 indicates tissue compression. Last, we computed the strain tensor ε (eq.6). A classic way of measuring strain, limited to a little amount of deformation, is with the right Cauchy- Green strain tensor defined as:

$$\varepsilon = \frac{1}{2}(F^T \cdot F - I) \tag{6}$$

where I is the undeformed triangular frame (identity matrix). ε is a matrix that provides information about the stretching and shearing of each mesh element. Looking at the diagonal values, we defined the main stretches and their direction as eigenvalues and eigenvectors for each mesh element. The anisotropy was determined by the ratio of its two main eigenvalues, which was denoted as θ. θ = 1 indicates isotropic deformation. θ > 1 indicates anisotropic deformation.

All these deformation measurements were translation and rotation invariant; therefore, they were not affected by relative movements of the embryo during image acquisition.

To minimize distortions related to isolated events, such as uncoordinated beating of cells during morphogenesis, we smoothed the deformation values within a geodesic distance up to 20 voxels.

The deformation pattern was calculated between two time-points and always mapped onto the shape of the last time considered.

**Concatenating individual motion profiles to compute the cumulative deformation.** To identify anatomical correspondences in the Atlas across the groups, we used motion profiles from different live images. SurfaceMaps represented individual motion profiles projected onto the Atlas. We established Atlas correspondences by aligning multiples SurfaceMaps. Equally staged SurfaceMaps had the same geometry, but different orientation and point density. To address this variability, we designated one SurfaceMap as the reference, known as eye-PC, and concatenated it with the hook-PC in a three-step process (Fig 4B). 1) We used the TMM-based rigid registration to align each hook-PC with the eye-PC [19,20]. This enabled us to overcome the differences in orientation between the two point-clouds; 2) We converted the registered hook-PC into binary masks and used a non-rigid registration algorithm to extract the T-transformation needed to morph each mask into the e27 SurfaceMaps mask [12]; 3) We employed a k-nearest neighbours algorithm (*knnsearch* function in MATLAB) to establish a face-to-face match between each hook-PC and the eye-PC. This step was crucial in addressing the differences in the density of points between the two point-clouds.

Referring to Fig 4C, we started the concatenation path at $Gr_4$ of embryo 27 (e27), using its SurfaceMap as our spatial reference. Embryos e02, e16, e124, e22, and e35 each had a staged frame in $Gr_4$. Through a face-to-face spatial mapping process, we projected the spatial reference (the e27 SurfaceMap) onto the SurfaceMaps of these other embryos (e02, e16, e124, e22, and e35) by identifying anatomical correspondences.

These correspondences between the eye-PC (e27) and the hook-PC (e02, e16, e124, e22, and e35) allowed us to indirectly determine the position of the e27 SurfaceMap in $Gr_3$.

To compare the deformation patterns of e31 and e05 in $Gr_3$, we used the anatomical correspondences in the e22 SurfaceMap as the eye-PC and applied face-to-face mapping between the SurfaceMaps of e31 and e05 and the eye-PC.

We used the same method to estimate the distribution of e27 SurfaceMap in $Gr_6$, this time using the e24 SurfaceMap for face-to-face mapping. To calculate the distribution in $Gr_8$, we fixed the anatomical correspondences in the e24 SurfaceMap as the eye-PC.

**Concatenating individual motion profiles to compute in-silico fate map.** To concatenate the different specimens and create a pseudo-cell tracking, we used their SurfaceMaps. As previously discussed, while the SurfaceMaps within the same group had similar shapes, they had different density of points.

To address this issue and establish correspondences between SurfaceMaps, we designated the SurfaceMaps of one embryo, e35, as the reference point cloud.

This decision was based on the fact that the live images of e35 covered a central sub-window of the developmental timeline. Therefore, we used e35 SurfaceMaps as the distribution of points to describe the motion profile of Atlas.

We followed the same concatenation approach as before, defining the eye-PC and the hookPC. We started by matching the SurfaceMap of e35 with that of e16 at $Gr_4$. Next, we used the motion profile of e16 to infer the position of the SurfaceMap of e35 at $Gr_3$. This became the new eye-PC to which the SurfaceMaps of e31 were anchored, and so on.

By combining the individual motion profiles, we created a Dynamic Atlas. The Dynamic Atlas maintained the original morphology of the static Atlas, but the positions of the mesh nodes were altered following the motion profile derived from the concatenation of the motion profiles of the live embryos.

**Computational resources.** The analyses were performed on a workstation equipped with an Intel(R) Xeon(R) E-2176M CPU operating at 2.70 GHz (2712 MHz), featuring 6 physical cores and 12 logical processors, 64 GB of RAM, and a NVIDIA Quadro P4200 GPU.

## Supporting information

**S1 Fig. Error estimation of motion profile computed for the resampled images.** The sequential error was estimated for the ground-truth points when the image was resized at 10%(A), 15%(B), 20%(C), and 25%(D). The error is expressed in μm. The red line indicates the averaged cell diameter of 20 μm. The yellow cubes indicate the voxel size. The registration time is half of the video time, according to the registration strategy adopted. The graphs represent the errors of 9584 tracked cells in the images of 9 embryos (e01, e02, e05, e06, e15, e16, e24, e26, e27).
(TIF)

**S2 Fig. Motion estimation error.** (A), The error made in continuous registration of landmarks belonging to different cardiac layers from images at 25% resolution. There is no statistically significant difference between the error made on myocardial cells, splanchnic mesodermal cells, and endodermal cells. (B), x-y-z decomposition of the continuous mean error, related to a 25% rescaled image (e01, e02, e05, e06, e15, e16, e24, e26, e27) is reported. (C) Number of landmarks in two-hours video interval for which the error was estimated.
(TIF)

**S3 Fig. Evaluation of h/w feature selected manually or computed by the motion profile.** The data are related to a single embryo (e01). (A, A'), In red the h and w lengths between landmarks taken manually on the shapes at each frame. In grey, the lengths h and w calculated automatically for each frame at the selected landmark at time t0 (registered point). (B), Comparison between the feature h/w obtained from the manual landmarks (red line) and the landmarks defined by the continuous model (gray line). (C), Percentage difference between the two features performed on 3 embryos (e01, e07, e27).
(TIF)

**S4 Fig. Concatenation paths defining fate map.** We evaluated the growth profile for the 10 zones between Gr2 to Gr8 for three different concatenation paths, shown in (A),(B),(C). We compare Gr2 to Gr8 as it represents the common stage range among the three different paths. The matching score between the growth profiles of the model and the one extracted from the live image was 62%, 65% and 81%, respectively. We chose the last concatenation path because it showed the highest match.
(TIF)

**S5 Fig. Validating the fate map at regional level.** The growth profile for the anatomical regions (excluding regions 1,5, and 10) were computed for the live-shapes and the Dynamic ATlas, with the corresponding heatmap. The two sorted growth trend matches at 92%.
(TIF)

**S1 Table. Dataset of live images.** The first column shows the list of embryos involved in our study; Genotype column reports the mouse allels (Nkx2.5GFP [[34], Polr2a–CreERT2 (RERT) [35], Nkx2.5cre/+ [11], Rosa26Rtdtomato + /- [36], Rosa26RmTmG [37], Tg(CBF:H2BVenus,+) [38]); TimeRes column lists the time resolution of the time-lapse[mm:ss]; VoxelRes column indicates the resolution in µm along the x,y,z axes; the last column (Tvideo) shows the total duration of the time-lapse[hh:mm]. e278 in grastrulation phase.
(TIF)

**S2 Table. Summary of tracked cells in different tissues.** We have indicated the number of frames for each video (Nframes), the duration of the video (T[hh:mm]), the number of totals tracked cells (NumCell). We have indicated which of these cells belong to the myocardium (Myo), how many to the mesoderm splachnic (Spl) and how many to the endoderm (Endo). In Totcells the total number of points, i.e., cells, for which the error is quantified. The total is given by the number of cells Numcells*Nframes. For e05 and e15, 1 and 4 endothelial cells were tracked respectively.
(TIF)

**S3 Table. Atlas staging system results.** Number of specimens used to build the Atlas Grs [9].
(TIF)

**S4 Table. Shapiro-Wilk test results for each Atlas group.**
(TIF)

**S5 Table. Comparison of Point Cloud rigid registration algorithms.** In each column, the rmse is reported for each embryo at each registration technique (ICP, CPD, TMM). We assessed the performance of the most popular methods for rigid point set registration: ICP [39] and the CPD [40] and TMM-based [20] algorithms. To this end, we evaluated how well the algorithms aligned a shape to a modified version of that shape. The altered shape underwent three modifications: 1) a random reduction of 30% of its points. 2) a rigid transformation as a rotation of 20° around zaxis; rotation of 70° around the x-axis; a rescale of 20% with respect to the original shape size. 3) a partial cut of the IFTs. The algorithms were tested to the first frame (t0) of five different embryos (e01, e02, e15, e16, e27) by calculating the root-mean-square error (rmse) between the original shape and the corresponding points in the altered one We found that the TMM-based algorithm performed better compared to ICP and CPD algorithms. So, we decided to implement the TMM-based algorithm as the first step in the spatial mapping pipeline.
(TIF)

**S6 Table. Staging system result.** The first column shows the Atlas group(Gr) associated with the live embryo shown in the second column. The last two columns show the video frames related to the rest state (frame(Gr-1)) and the deformed state (frame(Gr)), respectively.
(TIF)

**S7 Table. Validating the fate map at cellular level.**
(TIF)

**S1 Video. HT Morphogenesis as a continuum.** On the right, the live image of e01 captures the dynamic tissue movements during morphogenesis. On the left, the discretized model of the HT illustrates the motion profile.
(MP4)

**S2 Video. Spatio-temporal referencing of live images into the Atlas.** Frames 7, 19, and 29 of embryo e27 (grey shapes and grey mesh) are staged in Gr4, Gr5, and Gr6 of the Atlas. The staged frames are then mapped face-to-face into the Atlas (blue shapes). A yellow spot marks the corresponding anatomical position across the live images, live shapes, and the Atlas.
(MP4)

**S3 Video. In-silico fate map.** The dynamic Atlas is represented in its point cloud version. At the cellular level, a yellow spot tracks a hypothetical cell position throughout heart morphogenesis. At the tissue level, yellow spots illustrate the tissue deformation occurring during heart morphogenesis.
(MP4)

## Acknowledgments

We thank members of the Torres group for inspiring discussions and advice. We thank Peter Majer (Bitplane) for helpful advice and guidance on the work performed. We thank members of the Microscopy and Dynamic Imaging, Transgenesis, and Animal Facility CNIC units for excellent support.

## Author contributions

**Conceptualization:** Morena Raiola, Miguel Torres.

**Data curation:** Morena Raiola, Miguel Torres.

**Formal analysis:** Morena Raiola.

**Funding acquisition:** Miguel Torres.

**Investigation:** Isaac Esteban, Kenzo Ivanovitch, Miquel Sendra, Miguel Torres.

**Methodology:** Morena Raiola, Miquel Sendra.

**Project administration:** Miguel Torres.

**Software:** Morena Raiola.

**Supervision:** Miguel Torres.

**Writing – original draft:** Morena Raiola.

**Writing – review & editing:** Morena Raiola, Miguel Torres.

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
