## [Decision Letter · Decision Letter 0]

15 Aug 2025

PCOMPBIOL-D-25-01297

A method for analysing tissue motion and deformation during mammalian organogenesis

PLOS Computational Biology

Dear Dr. Torres,

Thank you for submitting your manuscript to PLOS Computational Biology. After careful consideration, we feel that it has merit but does not fully meet PLOS Computational Biology's publication criteria as it currently stands. Therefore, we invite you to submit a revised version of the manuscript that addresses the points raised during the review process.

Please submit your revised manuscript within 30 days Oct 15 2025 11:59PM. If you will need more time than this to complete your revisions, please reply to this message or contact the journal office at ploscompbiol@plos.org. Please include the following items when submitting your revised manuscript:

We look forward to receiving your revised manuscript.

Kind regards,

Juan Carlos del Alamo

Academic Editor

PLOS Computational Biology

Dimitrios Vavylonis

Section Editor

PLOS Computational Biology

**Journal Requirements:**

3) We notice that your supplementary Figures, and Tables are included in the manuscript file. Please remove them and upload them with the file type 'Supporting Information'. Please ensure that each Supporting Information file has a legend listed in the manuscript after the references list.

**Reviewers' comments:**

Reviewer's Responses to Questions

**Comments to the Authors:**

Reviewer #1: Authors present a powerful workflow to analyze live images of heart development in 3D+t. Each step presents a specific validation which is a good point. The paper integrates different techniques that have been developed in the last 10-15 years with more novel techniques.

Authors describe an atlas-based alignment using non-rigid registration. Please discuss how this affects to the deformation maps computation as the non-rigid registration can compensate inter-specimen variability of deformation patterns, growth and movements (including intercalation).

It is not clear if the motion estimation and deformation estimation are done before the template /atlas registration or after. Please clarify and discuss.

The optimization strategy does not look necessary as tracking methods already have tracking errors with maximal resolution. Please justify the need of reducing the resolution of the images. Is this a real-time application?

Authors should describe the interpretation of the two mechanical descriptors better: tissue growth rate (J) and tissue anisotropy (θ). It would be necessary to clearly define anisotropy of tissue dynamics. J and θ (as an angle) were already presented by Pastor-Escuredo et al iScience 2025 that presents deformation rates and cumulative deformation descriptors and ML analysis for fatemapping.

A further description and interpretation and scope of the descriptors is required. Also the projection/statistical matching of the descriptors into the model should be better described.

There is a lot of motion estimation description in the Results which can be valid, but there is little info on the statistical model. The statistical model should be further described as this is the most novel contribution of the methodology.

Authors should discuss better the limitations of doing cumulative rates onto a template/atlas instead of using trajectories (Lagrangian) as it has been done in other works. A better explanation of the dynamic atlas should be provided, it is a bit difficult to follow.

The discussion of the biological implications of the kinematic descriptors should be extended.

Also, the Introduction has little number of external references that would be desirable (Blanchard, Bellaiche, Heisenberg, Pastor-Escuredo, etc)

Apart from this, the manuscript seems a suitable contribution for the community.

Minor

“exhibits significant in” this phrase looks incomplete

Reviewer #2: In this manuscript, Raiola et al. present a computational method to analyse tissue motion and deformation in the mouse embryonic heart. This is a well done method with interesting innovations which is relevant to study organ morphogenesis.

There are a couple of relatively minor points that I think would benefit from further clarification:

On page 31, in the image preprocessing section, the authors mention resizing their 3D images to achieve isotropic resolution, but they don't specify what that resolution actually is. This could be linked to the optimization strategy section, where they aim to reduce the resolution of their 3D images to save computational time. One can assume that this downscaling occurs after achieving isotropic resolution, and that all axes are rescaled uniformly — but this is not clearly stated.

Additionally, Figure 2F doesn't help clarify this process. In fact, it seems to suggest that rescaling increases computational time, which contradicts the authors’ stated goal.

In that same section, they mention both reducing image resolution by 10–25%, and that computational time increases with higher resolutions (10–25%) — which creates some confusion. It seems like this part needs to be rephrased or clarified, as the current wording is somewhat contradictory.

**Have the authors made all data and (if applicable) computational code underlying the findings in their manuscript fully available?**

Reviewer #1: **No: ** I think that authors should make more data available on the statistical model and descriptors. It is not so evident to reproduce this from the code just with raw data.

Reviewer #2: Yes

PLOS authors have the option to publish the peer review history of their article (what does this mean? ). If published, this will include your full peer review and any attached files.

**Do you want your identity to be public for this peer review?** For information about this choice, including consent withdrawal, please see our Privacy Policy .

Reviewer #1: **Yes: ** David Pastor-Escuredo

Reviewer #2: No

**Figure resubmission:**
---

## [Editor Report · Decision Letter 1]

30 Sep 2025

Dear Dr Torres,

We are pleased to inform you that your manuscript 'A method for analysing tissue motion and deformation during mammalian organogenesis' has been provisionally accepted for publication in PLOS Computational Biology.

Best regards,

Juan Carlos del Alamo

Academic Editor

PLOS Computational Biology

Dimitrios Vavylonis

Section Editor

PLOS Computational Biology

---

## [Editor Report · Acceptance letter]

PCOMPBIOL-D-25-01297R1

A method for analysing tissue motion and deformation during mammalian organogenesis

Dear Dr Torres,

I am pleased to inform you that your manuscript has been formally accepted for publication in PLOS Computational Biology. Your manuscript is now with our production department and you will be notified of the publication date in due course.

With kind regards,

Anita Estes
